**Data Availability Statement:** All relevant data are within the manuscript and its Supporting information files.

# Network pharmacology and molecular docking reveal the effective substances and active mechanisms of Dalbergia Odoriferain protecting against ischemic stroke

**Kedi Liu**[1,2☯], **Xingru Tao**[1,2☯], **Jing Su**[1,2☯], **Fei Li**[3], **Fei Mu**[4], **Shi Zhao**[1,2], **Xinming Lu**[5], **Jing Li**[5], **Sha Chen**[5], **Taiwei Dong**[1], **Jialin Duan**[6]*, **Peifeng Wei**[7]*, **Miaomiao Xi**[2,7]*

1 College of Pharmacy, Shaanxi University of Chinese Medicine, Xianyang, Shaanxi, China, 2 TANK Medicinal Biology Institute of Xi'an, Xi'an, Shaanxi, China, 3 Department of Pharmacy, Tangdu Hospital, Fourth Military Medical University, Xi'an, Shaanxi, China, 4 Department of Pharmacy, Xijing Hospital, Fourth Military Medical University, Xi'an, Shaanxi, China, 5 YouYi Clinical Laboratories of Shaanxi, Xi'an, Shaanxi, China, 6 Institute of Medical Research, Northwestern Polytechnical University, Xi'an, China, 7 National Drug Clinical Trial Institute, The Second Affiliated Hospital, Shaanxi University of Chinese Medicine, Xianyang, Shaanxi, China

☯ These authors contributed equally to this work.
* dogson1989@163.com (JD); weipeifeng@163.com (PW); miaomiaoxi2014@163.com (MX)

## Abstract

Dalbergia Odorifera (DO) has been widely used for the treatment of cardiovascular and cerebrovascular diseasesinclinical. However, the effective substances and possible mechanisms of DO are still unclear. In this study, network pharmacology and molecular docking were used toelucidate the effective substances and active mechanisms of DO in treating ischemic stroke (IS). 544 DO-related targets from 29 bioactive components and 344 IS-related targets were collected, among them, 71 overlapping common targets were got. Enrichment analysis showed that 12 components were the possible bioactive components in DO, which regulating 9 important signaling pathways in 3 biological processes including 'oxidative stress' (KEGG:04151, KEGG:04068, KEGG:04915), 'inflammatory response'(-KEGG:04668, KEGG:04064) and 'vascular endothelial function regulation'(KEGG:04066, KEGG:04370). Among these, 5 bioactive components with degree≥20 among the 12 potential bioactive components were selected to be docked with the top5 core targets using Auto-dockVina software. According to the results of molecular docking, the binding sites of core target protein AKT1 and MOL002974, MOL002975, and MOL002914 were 9, 8, and 6, respectively, and they contained 2, 1, and 0 threonine residues, respectively. And some binding sites were consistent, which may be the reason for the similarities and differences between the docking results of the 3 core bioactive components. The results of *in vitro* experiments showed that OGD/R could inhibit cell survival and AKT phosphorylation which were reversed by the 3 core bioactive components. Among them, MOL002974 (butein) had a slightly better effect. Therefore, the protective effect of MOL002974 (butein) against cerebral ischemia was further evaluated in a rat model of middle cerebral artery occlusion (MCAO) by detecting neurological score, cerebral infarction volume and lactate dehydrogenase (LDH) level. The results indicated that MOL002974 (butein) could significantly improve

**Funding:** This study is supported by the National Natural Science Foundation of China(NO. 81470174 and 81903832), the Subject Innovation Team of the Second Affiliated Hospital of Shaanxi University of Chinese Medicine (NO.2020XKTD-A04) and the Outstanding Innovation Team of Shaanxi University of Chinese Medicine (No.ZYTD-04). The funders had no role in study design, data collection and analysis, decision to publish, or preparation of the manuscript.

**Competing interests:** The authors declare that they have no conflict of interest.

the neurological score of rats, decrease cerebral infarction volume, and inhibit the level of LDH in the cerebral tissue and serum in a dose-dependent manner. In conclusion, network pharmacology and molecular docking predicate the possible effective substances and mechanisms of DO in treating IS. And the results are verified by the *in vitro* and *in vivo* experiments. This research reveals the possible effective substances from DO and its active mechanisms for treating IS and provides a new direction for the secondary development of DO for treating IS.

## 1. Background

Stroke is the third leading death cause worldwide, which seriously threat human health. It is divided as ischemic stroke (IS) and hemorrhagic stroke (HS), of which IS accounts for more than 75% [1]. The possible mechanisms of IS includes oxidative stress, inflammation, apoptosis, energy metabolism disorders, etc [2, 3]. In clinical, thrombolytic agents are the commonly used drugs in treating IS, however the arrow time window limits its application [4]. Therefore, it is urgent to find novel drugs to treat IS.

Dalbergia Odorifera (DO; Chinese name, Jiangxiang) with the effects of promoting circulation and removing blood stasis, is an important composition of Huoxue Tongmai Capsule and Guanxin Danshen Capsule which were used in treating cardiovascular and cerebrovascular diseases [5]. The mainly chemical constituents of DO are flavonoids and volatile oil [6–8], which have effects of anti-oxidation [9–12], anti-inflammatory [13], and vascular endothelial function regulation [14–16]. In precious study, the effects of DO on IS had been well studied, however, the effective substances and active mechanisms were largely unknown to us, which limits the exploitation and application of this herb.

Herbs always contains many components (dozens or hundreds compounds), so it is difficult to study their effective substances and mechanisms using traditional methods. Network pharmacology is a novel method that combines systematic network analysis and pharmacology, which always used to clarify the synergistic effects and potential mechanisms of components-component networks, components-target networks, and targets-disease networks at the molecular level, so as to understand the interaction relationships among components, genes, proteins, and diseases [17]. Molecular docking is a computer-aided drug design technology, which uses computer technology to simulate the geometric structure and intermolecular interaction force of molecules through stoichiometric calculation methods to find the best binding mode of small molecule drugs and known structural macromolecules (proteins) [18]. Network pharmacology and molecular docking can study many chemicals and targets in the meantime, so they may provide possible research approaches for the TCM studying.

In this study, to illustrate the effective substances and possible mechanisms of DO, network pharmacology and molecular docking were used. The bioactive components from DO were screened, and the core targets of them were analyzed. Next, we constructed the multi-level interaction network of 'component-target-pathway', and explored the characteristics of the combination between chemicals and targets. Finally, *in vitro* experiments were used to verify these results. The study flowchart of network pharmacology and molecular docking is displayed in Fig 1.

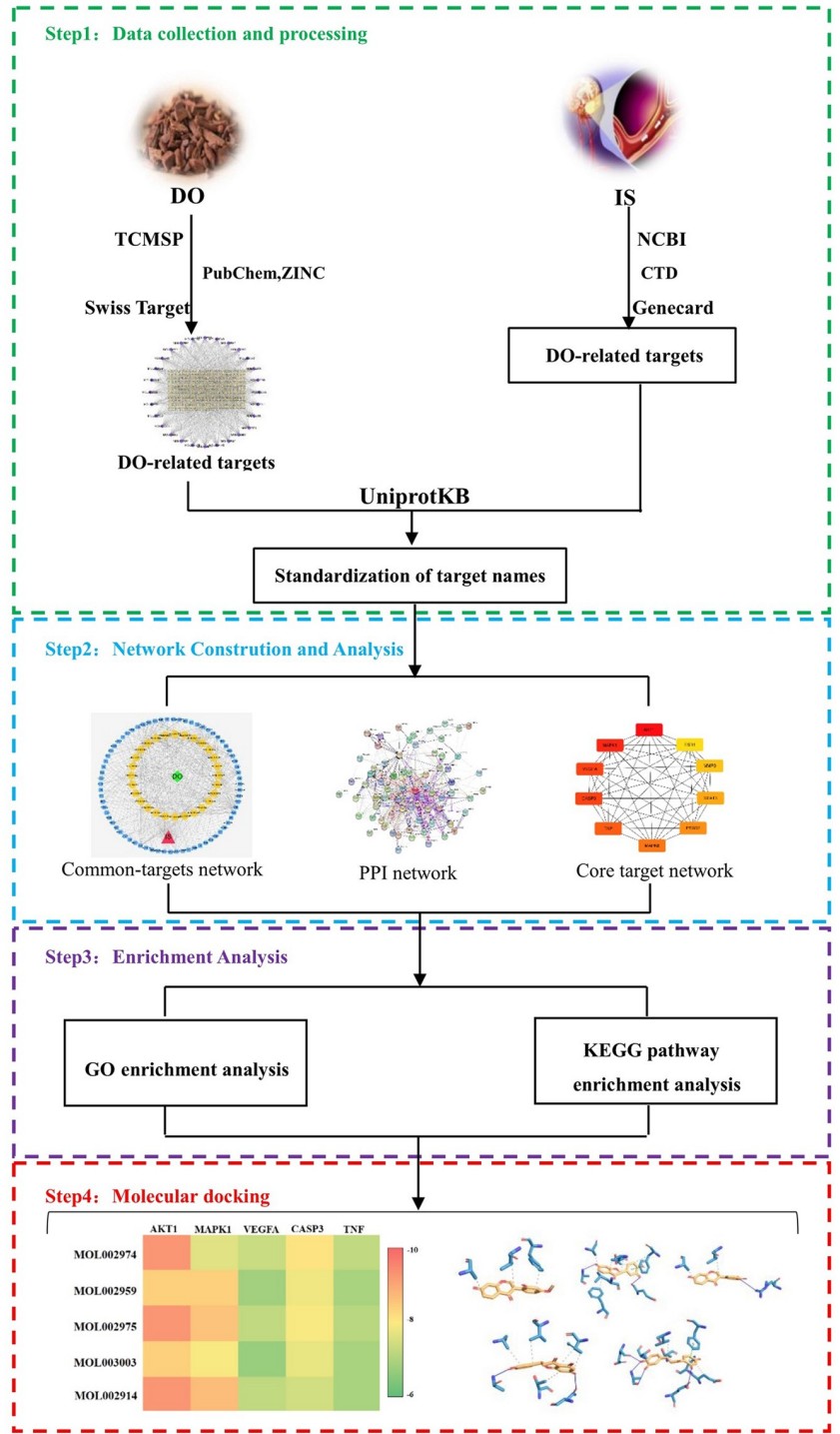

**Fig 1. Study flowchart of network pharmacology and molecular docking.**

## 2. Methods

### 2.1. Data collection and processing

**2.1.1. Composite components of DO.**   Data of the compounds of DO were mainly collected from the natural product databases for Chinese herbal medicine: Traditional Chinese Medicine Systems Pharmacology Database and Analysis Platform (TCMSP, http://ibts.hkbu.edu.hk/LSP/tcmsp.php).

**2.1.2. Screening of bioactive components.**   TCM is mainly used by oral administration. Therefore, the bioactive components of DO were screened by 2 main parameters affecting gastrointestinal absorption: oral bioavailability(OB)$\geq$40% and drug-likeness(DL)$\geq$0.18.

**2.1.3. Target prediction of bioactive components.**   Swiss Target Prediction (http://www.swisstargetprediction.ch/) is a web server based on 2D and 3D similarity measurement of known ligands, which can accurately predict the target of bioactive components. First, we obtained the molecular structure of bioactive components in DO and the structural formulas 'canonical smiles' and 'sdf' from PubChem (https://pubchem.ncbi.nlm.nih.gov/) and ZINC databases (http://zinc15.docking.org). Then we used the species as 'Homo sapiens' to predict the target through the Swiss Target Database, and used UniProt knowledge database (https://sparql.uniprot.org/) to standardize the target name and removed the duplicate target.

**2.1.4. IS-related targets.**   We searched the keyword 'stroke' in the GeneCards database (Gene, https://www.genecards.org/) and the National Centre for Biotechnology Information Gene (NCBI, https://www.ncbi.nlm.nih.gov/gene/). Also, we searched keywords such as 'stroke' and 'cerebral infarction' in the Comparative Toxicogenomics Database (CTD, http://ctdbase.org/). Then, the results of the 3 databases were summarized and the duplicates were deleted.

### 2.2. Network construction

**2.2.1. Common-target network construction.**   We screened the common targets of DO bioactive components and IS, and used Cytoscape (http://www.cytoscape.org) to build a common target network.

**2.2.2. PPI network construction.**   PPI network as a new drug research method can be used to clarify the relationship between the predicted targets and other human proteins. The String (https://string-db.org/) is a database for searching protein interactions, which provides information on protein prediction and experimental interactions. The PPI network of common targets of IS-related and DO-related targets were constructed by String database and the visualization of the PPI network was achieved by Cytoscape 3.7.1 software.

### 2.3. Core target prediction

A topological analysis was performed on the network to screen out the core targets of top10 by using the cytoHubba. The more darker of the color means the more targets connected to it and the closer connection.

### 2.4. Screening the potential bioactive components

A topological analysis was performed on the network by using Analyze Network to sort according to the size of the node Degree, and to screen out the potential bioactive components according to 'Degree> Median'.

## 2.5. Enrichment analysis

We used 'P≤0.05 using the Bonferroni correction' as the screening condition to perform GO enrichment analysis on the network through the DAVID database. Then, with 'P≤0.05' as the screening condition, we used the ClueGO software to analyze the KEGG channel enrichment of the network.

## 2.6. Molecular docking

We downloaded the target protein structures from the RSCB PDB database (http://www.rcsb.org/), and used PyMOL software to remove the crystalline water and other small molecules of each protein structure, and saved it as pdb format. Then we imported the structure files into the AutodockTool 1.5.6 program, added the atomic charge, and saved it as pdbqt format after adding hydrogenation. Next, the mol2 format file of the bioactive components were imported into the AutodockTool1.5.6 program, and added the atomic charge, then saved it as the pdbqt format as the docking ligands. Autodock vina software was used to simulate molecular docking to determine the binding affinity of the target proteins and the potential bioactive components [19]. The binding affinities between these bioactive components and the target proteins were used as the evaluation criteria. The smaller the binding affinity, the better the docking is.

## 2.7. Experimental verification

**2.7.1 Materials.** MOL002974 (butein), MOL002975 (butin), MOL002914 (Eriodyctiol) were purchased from Shanghai Yuanye Biotechnology Co., Ltd. (HPLC≥98%). Edaravone injection was purchased from Jilin Boda Pharmaceutical Co., Ltd.; PC12 cells were donated by the Department of Toxicology, Department of Preventive Medicine, Air Force Military Medical University. Sprague-Dawley rats (weight 250.0±5.0g) were provided by the Experimental Animal Center of the Fourth Military Medical University, and the production license number is SCXK (Shaanxi) 2019–001. All animals followed the relevant regulations of experimental animal ethics and passed the animal experiment ethics review of Shaanxi University of Traditional Chinese Medicine. Rats were reared in separate cages, with strict control of temperature (22±2˚C), humidity (55%-75%), light (12-hour cycle) and other feeding conditions. Primary antibodies against GAPDH were purchased from Cell Signaling Technology (Danvers, MA, USA). Primary antibodies against AKT and p-AKT were purchased from ProteinTech Group, Inc. (Rosemont, IL, USA). Secondary antibody was bought from Abbkine (California, USA).

**2.7.2. Cell culture and treatment.** PC12 cells were cultured by a DMEM high-glycemic medium containing 10% FBS and 1% bi-antibody in a constant temperature incubator with 37˚C and 5% $CO_2$. For oxygen-glucose deprivation, PC12 cells were cultured with glucose-free DMEM medium in an anaerobic incubator (95% $N_2$, 5% $CO_2$) at 37˚C for 3 hours. For reperfusion, the culture medium was replaced with complete medium and cultured for another 24 hours, these process was defined as oxygen-glucose deprivation/reperfusion model (OGD/R). The control group was cultured in normal DMEM medium under normal culture conditions. After 3 hours of an aerobic incubation, the MOL002974 treatment groups were replaced with a complete medium containing 0.5, 1, and 2 μmol/L MOL002974 and cultured for 24 hours, the MOL002975 and MOL002914 treatment groups were replaced with a complete medium containing1, 2, and 4 μmol/L MOL002975, MOL002914 and cultured for 24 hours.

**2.7.3. Cell viability analysis.** PC12 cells were seeded in a 96-well plate at a cell density of $5×10^4$ cells/well. After the cells adhered, they were grouped according to "2.7.2". After the culture, the medium was replaced with CCK8 medium and incubated in a 37˚C, 5% $CO_2$ cell incubator for 4 hours, and the absorbance value of each well was measured at a wavelength of

450 nm using a microplate reader.

$$\begin{aligned}&\text{Survival rate}(\%) \\ &= (\text{OD}_{\text{addition group}} - \text{OD}_{\text{blank group}})/(\text{OD}_{\text{normal control group}} - \text{OD}_{\text{blank group}}) \times 100\%.\end{aligned}$$

**2.7.4. Western blot analysis.** First, the cells were lysed with a pre-cooled RIPA lysate buffer containing protease inhibitors. The cell lysates were centrifuged at 12000 g for 10 min in 4˚C, and the concentration of protein were determined by BCA method. The same amount of protein (20ug) was isolated with 10% SDS-PAGE and transferred onto the PVDF membrane. At room temperature, the membrane were blocked by 2h in TBST with 5% milk and incubated overnight at 4˚C with primary antibodies. Then the membranes were incubated with secondary antibody at 37˚C for 2h. The blots were visualized by ECL reagents, the bands were scanned and analyzed by quantitative-image analysis software.

**2.7.5. Animals grouping and treatment.** The rats were randomly divided into 7 groups, with six rats in each. Sham rats received saline by intraperitoneal injection (20ml/kg) once a day for 3 consecutive days. Model rats received saline by intraperitoneal injection (20ml/kg) once a day for 3 consecutive days. Edaravone group rats (Eda group) received edaravone injection by intraperitoneal injection (3mg/kg) once a day for 3 consecutive days. Solvent control group rats (DMSO group) received DMSO by intraperitoneal injection (1.9mg/kg) once a day for 3 consecutive days. Butein high-dose group rats (butein-8 group) received butein by intraperitoneal injection (8mg/kg) once a day for 3 consecutive days. Butein medium-dose group rats (butein-4 group) received butein by intraperitoneal injection (4mg/kg) once a day for 3 consecutive days. Butein low-dose group rats (butein-2 group) received butein by intraperitoneal injection (2mg/kg) once a day for 3 consecutive days.

**2.7.6. Samples collection and processing.** 24 hours after the last injection of butein, the rats were anesthetized by intraperitoneal injection of 4% pentobarbital sodium.1ml of blood was taken from the abdominal aorta, left standing at room temperature for 30 minutes, centrifuged at 3500g for 3 minutes, and the supernatant (serum) was taken for LDH testing. The whole brain was removed, washed with pre-cooled 1×PBS, and dried with filter paper 3 whole brain of each group was used for cerebral infarction volume detection and 3 whole brain of each group was used for LDH level testing.

**2.7.7. Neurofunctional scores analysis.** After 24 h of reperfusion, neurological deficits were evaluated by a blinded observer with a 5-point-scale scoring system as described previously. 0 = no obvious neurological deficit; 1 = inability to extend the contralateral forelimb; 2 = circle to the opposite side of ischemia; 3 = unable to bear the weight of the contralateral side; 4 = no voluntary movement or disturbance of consciousness.

**2.7.8. Cerebral infarct volume analysis.** After neurological assessment, the rat was decapitated and the cerebral was taken out for the infarct volume measurement. The whole brain was sliced into uniform coronal slices, each slice 2 mm thick. The sections were stained with 1% 2,3,5-triphenyltetrazolium chloride (TTC), kept at 37˚C for 10 minutes, and fixed in 4% paraformaldehyde buffer. For analysis, the slices were photographed with a digital camera. A computerized image analysis system was used to determine the infarct area of each slice. Infarct volume was expressed as percentages of contralateral hemispheric volume.

**2.7.9. Lactate dehydrogenase (LDH) levels analysis.** The levels of LDH in serum and cerebral tissue were detected by using LDH assay kit following the manufacturer's instruction. The data were measured by a microplate reader at 440 nm.

## 2.8. Statistical analysis

SPSS20.0 software was used to perform statistical analyses. The results were shown as $\bar{X} \pm SD$. ANOVA followed by LSD-t test were used for mean comparison between groups, with $P<0.05$ indicating statistical significant.

# 3. Results

## 3.1. DO component-target network

98 components were collected in DO based on the TCMSP database, and 28 bioactive components were selected with the screening conditions of OB≥40% and DL≥0.18 (Fig 2). In previous and our preliminary studies, butein has protective effects against IS, so butein was selected in the further studies [20].

Next, we used the Swiss Target Prediction database to predict the target of 29 monomer structures, set the species to 'Homo sapiens', removed the duplicates and used UniProtKB to standardize the target name. A total of 544 targets were obtained. The component-target network was constructed by using Cytoscape software, which contained 572 nodes and 2772 edges (Fig 3).

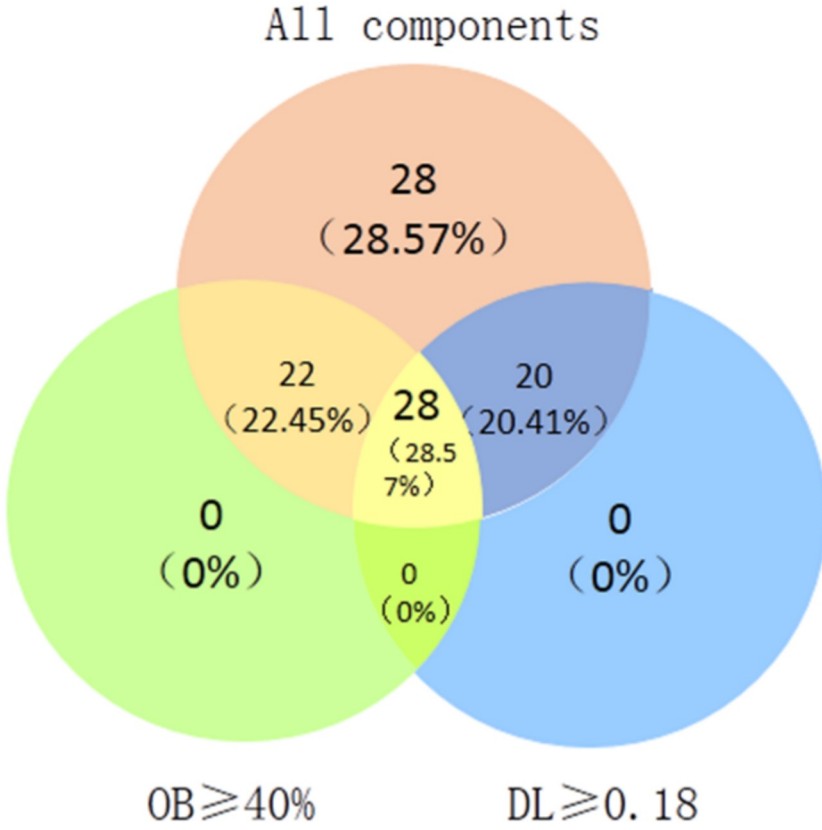

**Fig 2. DO component-target network.** Venn diagram: 98 components (pink section), and 28 bioactive components screened by two ADME-related models (green section stands for the components of OB≥40%, blue section stands for DL≥0.18).

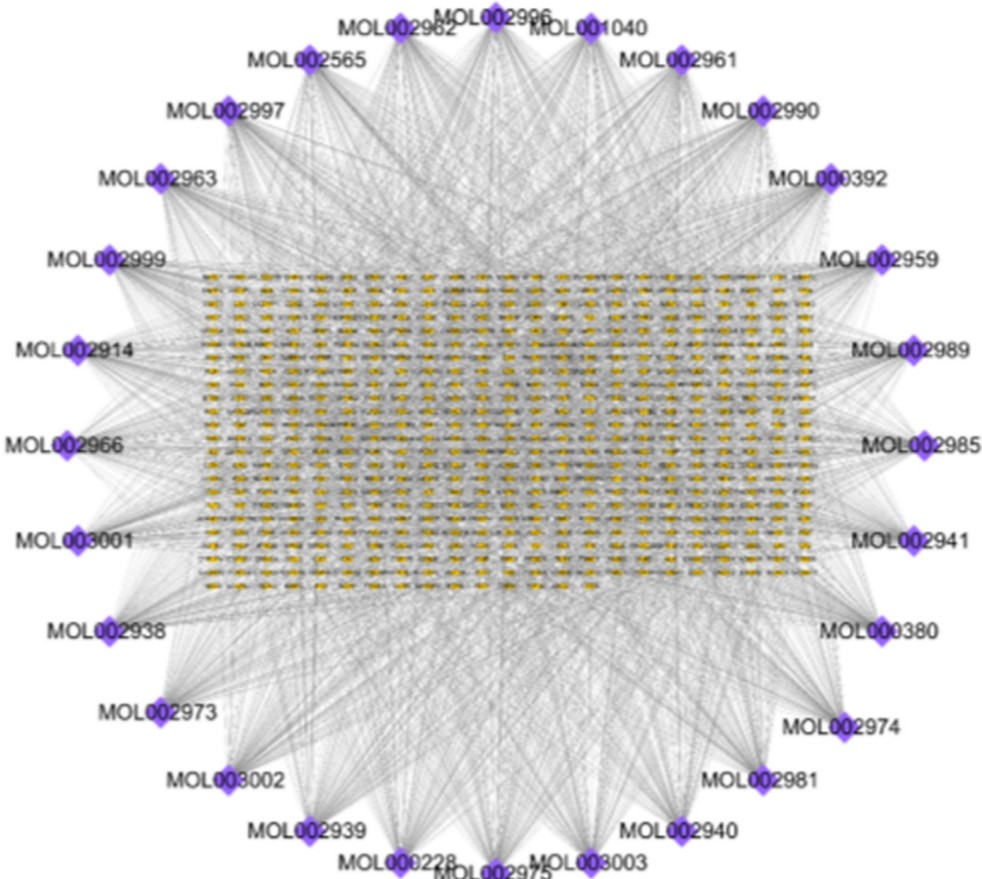

**Fig 3. Construction of DO component-targets visual network, including 572 nodes and 2772 edges.** Blue nodes stand for bioactive components from DO, yellow nodes stand for targets.

## 3.2. Common-target network

The occurrence and development of IS involves the co-regulation of multiple genes. We screened 126, 196, and 129 targets from 3 databases (NCBI, CTD, and GeneCards), and collected a total of 344 targets related to IS. Using Venny online drawing tool, 544 monomer component targets and 344 disease targets were intersected, and a total of 71 common targets were obtained (Fig 4A, Table 1). Then, we built a complex network based on the interactions among bioactive components, targets and the disease (IS) by using the Cytoscape software and gained a network which was made up of 101 nodes and 498 edges (Fig 4B). The importance of the node in terms of degree and intermediary degree was reflected using Cytoscape software for topological analysis on the obtained network graph. The values of nodes and betweenness centrality were commonly used to describe the importance of network nodes.

## 3.3. PPI network construction

In this study, we used the string tool to acquire PPI network for the 71 overlapped targets. With a combined score greater than 0.4 and 'Homo sapines'as selecting criterions, the network of PPI consisted of 71 nodes and 815 edges (Fig 5). Each node represents the relevant gene, the edge means line thickness indicates the strength of data support.

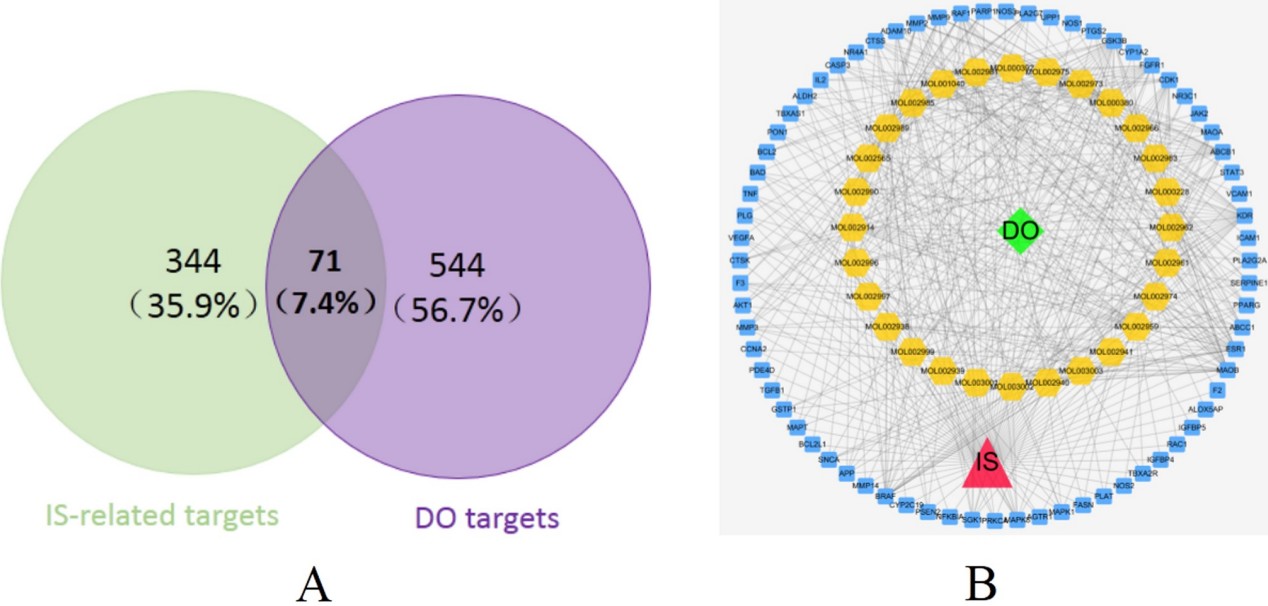

**Fig 4. Common-targets network.** (A) 71 targets are common to DO and IS. (B) Common-target network, including 101 nodes and 498 edges. Yellow nodes stand for bioactive components from DO, blue nodes stand for targets.

### 3.4. Core target network

Top10 core targets in the PPI network were obtained by cytoHubba plug-ins, followed by AKT1, MAPK1, VEGFA, CASP3, TNF, MAPK8, PTGS2, STAT3, MMP9 and ESR1 (Fig 6). The darker the node color, the more targets are connected to it and the greater its importance is of the possible role in the occurrence and development of IS.

### 3.5. Screening of potential bioactive components

We screened 12 potential bioactive components using the 'Degree>median' and 'Degree>14' as screening criteria (Fig 7).

We analyzed the structure of 12 compounds screened by network pharmacology and found that these were flavonoids, including dihydroflavonoids (1. MOL002914, 4. MOL002975, 6. MOL000228, 7. MOL001040, 8. MOL002989, 10. MOL002999, 11. MOL002938, 12. MOL002985), isoflavones (2. MOL002974, 3. MOL002959, 5. MOL003003, 9. MOL000392). Studies have shown that the flavonoids of DO have anti-inflammatory, anti-oxidant, anti-thrombotic, anti-platelet aggregation, anti-tumor and other pharmacological effects.

### 3.6. Enrichment analysis

GO functional enrichment analysis and KEGG pathway enrichment in DAVID resulted in 182 entries ($P<0.05$) and 107 signaling pathways ($P<0.05$). Among them, GO functional enrichment analysis included 148 biological processes (BPs), 16 cell compositions (CCs), and 18 molecular functions (MFs), accounting for 81%, 9%, and 10%, respectively. The biological processes mainly involved lipopolysaccharide-mediated signaling pathway, positive regulation of protein phosphorylation, cell response to organic cyclic compound, cellular response to vascular endothelial growth factor stimulus, angiogenesis, apoptotic process, positive regulation of nitric oxide biosynthesis process, response to drug and response to hypoxia. The cell

**Table 1. 71 common targets among the targets related to DO and the targets related to IS.**

| Number | Uniprot ID | Target name | Target name abbreviation |
|---|---|---|---|
| 1 | P27338 | Monoamine oxidase B | MAOB |
| 2 | P03372 | Estrogen receptor alpha | ESR1 |
| 3 | P33527 | Multidrug resistance-associated protein 1 | ABCC1 |
| 4 | P37231 | Peroxisome proliferator-activated receptor gamma | PPARG |
| 5 | P05121 | Plasminogen activator inhibitor-1 | SERPINE1 |
| 6 | P14555 | Phospholipase A2 group IIA | PLA2G2A |
| 7 | P35968 | Vascular endothelial growth factor receptor 2 | KDR |
| 8 | P40763 | Signal transducer and activator of transcription 3 | STAT3 |
| 9 | P08183 | ATP-dependent translocase ABCB1 | ABCB1 |
| 10 | P21397 | Monoamine oxidase A | MAOA |
| 11 | O60674 | Tyrosine-protein kinase JAK2 | JAK2 |
| 12 | P06493 | Cyclin-dependent kinase 1 | CDK1 |
| 13 | P11362 | Fibroblast growth factor receptor 1 | FGFR1 |
| 14 | P49841 | Glycogen synthase kinase-3 beta | GSK3B |
| 15 | P35354 | Prostaglandin G/H synthase 2 | PTGS2 |
| 16 | Q16831 | Uridine phosphorylase 1 (by homology) | UPP1 |
| 17 | Q13093 | Platelet-activating factor acetylhydrolase | PLA2G7 |
| 18 | P29474 | Nitric-oxide synthase, endothelial (by homology) | NOS3 |
| 19 | P37840 | Alpha-synuclein | SNCA |
| 20 | P05067 | Amyloid-beta precursor protein | APP |
| 21 | P13726 | Tissue factor | F3 |
| 22 | P10636 | Microtubule-associated protein tau | MAPT |
| 23 | P35228 | Nitric oxide synthase, inducible | NOS2 |
| 24 | P05091 | Aldehyde dehydrogenase2 | ALDH2 |
| 25 | P08254 | Matrix metalloproteinase 3 | MMP3 |
| 26 | P04049 | Serine/threonine-protein kinase RAF | RAF1 |
| 27 | P15056 | Serine/threonine-protein kinase B-raf | BRAF |
| 28 | P08253 | Matrix metalloproteinase 2 | MMP2 |
| 29 | P63000 | Ras-related C3 botulinum toxin substrate 1 | RAC1 |
| 30 | P20292 | Arachidonate 5-lipoxygenase-activating protein | ALOX5AP |
| 31 | P00734 | Prothrombin | F2 |
| 32 | P50281 | Matrix metalloproteinase 14 | MMP14 |
| 33 | P15692 | Vascular endothelial growth factor A | VEGFA |
| 34 | P43235 | Cathepsin K | CTSK |
| 35 | P31749 | Serine/threonine-protein kinase AKT | AKT1 |
| 36 | P10415 | Apoptosis regulator Bcl-2 | BCL2 |
| 37 | P14780 | Matrix metalloproteinase 9 | MMP9 |
| 38 | P09874 | Poly [ADP-ribose] polymerase-1 | PARP1 |
| 39 | O14672 | ADAM10 | ADAM10 |
| 40 | P25774 | Cathepsin S | CTSS |
| 41 | P22736 | Nuclear receptor subfamily 4 group A member 1 | NR4A1 |
| 42 | P42574 | Caspase-3 | CASP3 |
| 43 | P60568 | Interleukin-2 | IL2 |
| 44 | P24557 | Thromboxane-A synthase | TBXAS1 |
| 45 | P27169 | Serum paraoxonase/arylesterase 1 | PON1 |
| 46 | Q92934 | Bcl2-antagonist of cell death | BAD |
| 47 | P01375 | TNF-alpha | TNF-α |

(*Continued*)

**Table 1.** (Continued)

| Number | Uniprot ID | Target name | Target name abbreviation |
|---|---|---|---|
| 48 | P00747 | Plasminogen | PLG |
| 49 | Q08499 | Phosphodiesterase 4D | PDE4D |
| 50 | P09211 | Glutathione S-transferase Pi | GSTP1 |
| 51 | P33261 | Cytochrome P450 2C19 | CYP2C19 |
| 52 | P49810 | Presenilin-2 | PSEN2 |
| 53 | P25963 | NF-kappa-B inhibitor alpha | NFKBIA |
| 54 | O00141 | Serine/threonine-protein kinase Sgk1 | SGK1 |
| 55 | P17252 | Protein kinase C alpha type | PRKCA |
| 56 | P45983 | c-Jun N-terminal kinase 1 | MAPK8 |
| 57 | P30556 | Type-1 angiotensin II receptor (by homology) | AGTR1 |
| 58 | P28482 | MAP kinase ERK2 | MAPK1 |
| 59 | P49327 | Fatty acid synthase | FASN |
| 60 | P00750 | Tissue-type plasminogen activator | PLAT |
| 61 | P21731 | Thromboxane A2 receptor | TBXA2R |
| 62 | P22692 | Insulin-like growth factor binding protein 4 | IGFBP4 |
| 63 | P24593 | Insulin-like growth factor binding protein 5 | IGFBP5 |
| 64 | P05362 | Intercellular adhesion molecule-1 | ICAM1 |
| 65 | P19320 | Vascular cell adhesion protein 1 | VCAM1 |
| 66 | P04150 | Glucocorticoid receptor | NR3C1 |
| 67 | P05177 | Cytochrome P450 1A2 | CYP1A2 |
| 68 | P29475 | Nitric-oxide synthase, brain | NOS1 |
| 69 | P01137 | Transforming growth factor beta-1 | TGFB1 |
| 70 | Q07817 | Apoptosis regulator Bcl-X | BCL2L1 |
| 71 | P20248 | CDK2/Cyclin A | CCNA2 |

compositions mainly involved extracellular space, platelet alpha granule lumen, extracellular region, proteinaceous extracellular matrix, plasma membrane and cell surface. The molecular functions mainly involved protein binding, enzyme binding, serine-type endopeptidase activity, metallopeptidase activity and protease binding.

According to IS pathogenesis, these biological processes can be divided into 3 parts, including oxidative stress (GO:0071407, GO:0032355, GO:0045429, GO:0051926, GO:0070374, GO:0043066, KEGG:04151, KEGG:04068, KEGG:04915, KEGG:04010, KEGG:04014), inflammatory response (GO:0043066, GO:0071222, GO:0071260, KEGG:04668, KEGG:04010, KEGG:04014, KEGG:04064) and regulation of vascular endothelial function (GO:0043536, GO: 0045766, GO:0001666, GO:0071456, GO:0043066, KEGG:04370, KEGG: 04066) (Table 2).

## 3.7. Molecular docking

Molecular docking is one of the most important and commonly used methods for comparing the biological activity of molecules on enzymes, and the most important parameter of molecular docking is affinity. The molecule with the lowest value of this parameter has the highest biological activity. To further screened the core bioactive components of DO affecting on IS, we tested the affinity of 5 potential bioactive components and the following top5 core target proteins: AKT1 (PDB: 3CQU), MAPK1 (PDB: 5K4I), VEGFA (PDB: 3BDY), CASP3 (PDB: 3H0E) and TNF (PDB: 4TWT), respectively (Fig 8). The results showed that all the 5 potential

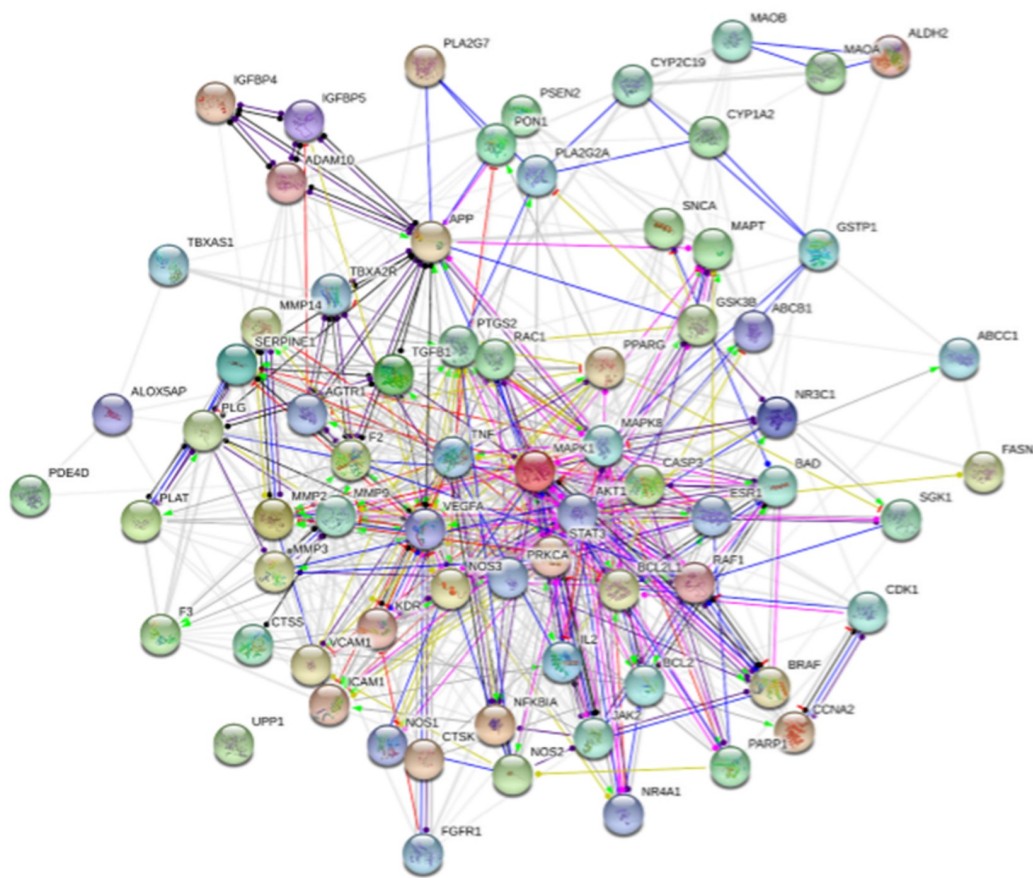

**Fig 5. Protein–protein interaction (PPI) networks of bioactive components of DO for the treatment of IS.** Each node represents the relevant gene, the edge means line thickness indicates the strength of data support.

bioactive components could stably bind to the top5 targets and strongly bind to the target protein AKT1, so we focused on analyzing the binding sites of MOL002974, MOL002914 and MOL002975 with AKT1 (Fig 9). According to the ligand-protein interaction after molecular docking, it could be found that the binding of each small molecule to the target protein mainly depended on hydrophobic interaction and hydrogen bonding (Table 3). MOL002974 combined with residues such as THR211, THR291, GLU278, PHE161, VAL164 on the AKT1 binding site to form a hydrophobic interaction, and combined with the residues of ASN279, GLU228, ALA230 and GLU234 to form hydrogen bonds (Fig 10A). MOL002975 combined with the residues of THR291, PHE161, ALA177, VAL164 and LEU156 on the AKT1 binding site to form a hydrophobic interaction, and combined with the residues of ALA230 and GLU278 to form hydrogen bonds (Fig 10B). MOL002914 combined with the residues such as THE291, ALA177, VAL164, PHE161 on the AKT1 binding site to form a hydrophobic interaction, and combined with the residues of ALA230 and ASN279 to form hydrogen bonds (Fig 10C). MOL2959 binded with the residues of ALA177 and VAL164 on the AKT1 binding site to form a hydrophobic interaction, and combined with the residues of ARG4 to form hydrogen bonds (Fig 10D). MOL003003 binded with the residues of ALA177, VAL164 and PHE161 on the AKT1 binding site to form a hydrophobic interaction (Fig 10E).

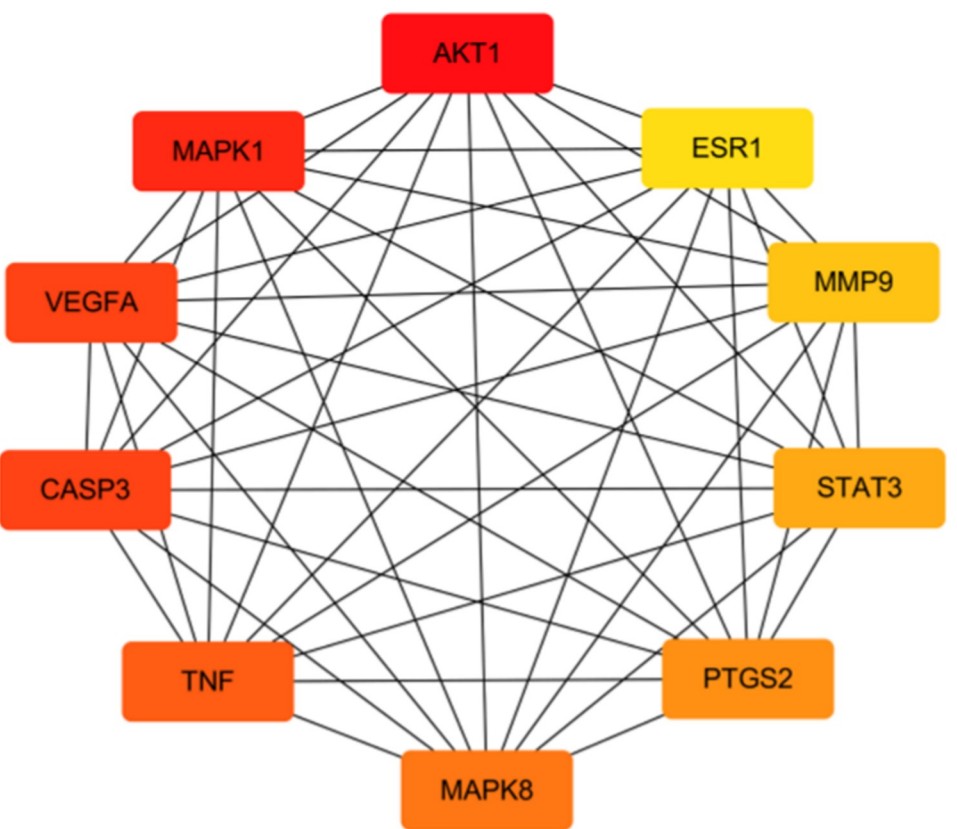

**Fig 6. Hub top 10 genes in PPI network, the darker the color, the higher the score.**

### 3.8. Experimental verification

**3.8.1. Cell viability analysis.** Compared with the Control group, the cell survival rate of the OGD/R group was significantly reduced ($P<0.01$), suggesting that OGD/R caused PC12 cell damage. Compared with the OGD/R group, the MOL002974 low-dose groups and MOL002975 low-dose groups had a tendency to increase cell survival, but there was no significant difference. The survival rate of cells in the medium-dose group ($P<0.05$) and high-dose group ($P<0.01$) gradually increased (Fig 11A and 11B). MOL002914 low-dose group and medium-dose group had a tendency to increase cell survival rate, but there was no significant difference. The high-dose group ($P<0.05$) increased cell survival rates (Fig 11C). It is suggested that MOL002974, MOL002975, and MOL002914 could promote PC12 cells survived in a dose-dependent manner. The effects of the 3 core bioactive components on the survival rate of PC12 cells were further compared at the same concentration, and the results showed that MOL002974 had a slightly better effect on improving the survival rate of cells (Fig 11D).

**3.8.2. Western blot analysis.** Based on the results of network pharmacology and molecular docking, we used western blotting to verify the regulation of the core bioactive components MOL002974 (Fig 12A), MOL002975 (Fig 12B) and MOL002914 (Fig 12C) on the target of AKT in PC12 cells. The results showed that, compared with the control group (no treatment), the OGD/R group inhibited the phosphorylation of AKT (p<0.01). However, 3 core bioactive component treatments significantly up-regulated the phosphorylation of AKT in a dose dependent manner (p<0.01), respectively. In addition, 3 core bioactive components could up-

| Number | MolID | Component | Degree | Molecular formula | Structure |
|---|---|---|---|---|---|
| 1 | MOL002914 | Eriodyctiol (flavanone) | 24 | $C_{15}H_{12}O_6$ | |
| 2 | MOL002974 | Butein | 23 | $C_{15}H_{12}O_5$ | |
| 3 | MOL002959 | 3'-Methoxydaidzein | 22 | $C_{16}H_{12}O_5$ | |
| 4 | MOL002975 | butin | 21 | $C_{15}H_{12}O_5$ | |
| 5 | MOL003003 | Xenognosin B | 20 | $C_{16}H_{11}O_5$ | |
| 6 | MOL000228 | (2R)-7-hydroxy-5-methoxy-2-phenylchroman-4-one | 19 | $C_{16}H_{14}O_4$ | |
| 7 | MOL001040 | (2R)-5,7-dihydroxy-2-(4-hydroxyphenyl)chroman-4-one | 19 | $C_{15}H_{12}O_5$ | |
| 8 | MOL002989 | 4-Hydroxyhomopterocarpin | 17 | $C_{17}H_{16}O_5$ | |
| 9 | MOL000392 | formononetin | 16 | $C_{16}H_{12}O_4$ | |
| 10 | MOL002999 | Sativanone | 16 | $C_{17}H_{16}O_5$ | |
| 11 | MOL002938 | (3R)-4'-Methoxy-2',3,7-trihydroxyisoflavanone | 15 | $C_{16}H_{14}O_6$ | |
| 12 | MOL002985 | isoduartin | 15 | $C_{18}H_{18}O_6$ | |

**Fig 7. Potential bioactive components of DO treating IS.**

regulate AKT phosphorylation under the same conditions, and MOL002974 had a slightly better effect (Fig 12D).

**3.8.3. Neurofunctional scores analysis.** The rats in each group were scored for neurological function before being sacrificed. Compared with the Sham group, the neurofunctional scores of rats in the MCAO group increased significantly ($P<0.01$). Compared with the MCAO group, butein low-dose ($P<0.05$), medium-dose ($P<0.01$), high-dose group ($P<0.01$) and Eda group ($P<0.01$) significantly reduced the neurological scores of rats, suggesting butein can reduce cerebral damage caused by CI/R in a dose-dependent manner (Fig 13).

**3.8.4. Cerebral infarct volume analysis.** Compared with the Sham group, the cerebral infarction volume of rats in the MCAO group increased significantly ($P<0.01$). Compared with the MCAO group, butein medium-dose ($P<0.05$), high-dose group ($P<0.05$) and Eda

**Table 2. Functions of 71 common targets based on GO and KEGG pathway analysis through DAVID and ClueGO.**

| Classification | ID | Term |
|---|---|---|
| Oxidative stress | GO:0071407 | cellular response to organic cyclic compound |
| | GO:0032355 | response to estradiol |
| | GO:0045429 | positive regulation of nitric oxide biosynthetic process |
| | GO:0051926 | negative regulation of calcium ion transport |
| | GO:0070374 | positive regulation of ERK1 and ERK2 cascade |
| | GO:0043066 | negative regulation of apoptotic process |
| | KEGG:04151 | PI3K-Akt signaling pathway |
| | KEGG:04068 | FoxO signaling pathway |
| | KEGG:04915 | Estrogen signaling pathway |
| | KEGG:04010 | MAPK signaling pathway |
| | KEGG:04014 | Ras signaling pathway |
| Inflammatory response | GO:0031663 | Lipopolysaccharide-mediated signaling pathway |
| | GO: 0071222 | cellular response to lipopolysaccharide |
| | GO:0071260 | Cellular response to mechanical stimulus |
| | KEGG:04668 | TNF signaling pathway |
| | KEGG:04064 | NF-κB signaling pathway |
| | KEGG:04010 | MAPK signaling pathway |
| | KEGG:04014 | Ras signaling pathway |
| Vascular endothelial function regulation | GO: 0043536 | positive regulation of blood vessel endothelial cell migration |
| | GO: 0045766 | positive regulation of angiogenesis |
| | GO:0001666 | Response to hypoxia |
| | GO:0071456 | Cellular response to hypoxia |
| | GO:0043066 | Negative regulation of apoptotic process |
| | KEGG:04066 | HIF-1 signaling pathway |
| | KEGG:04370 | VEGF signaling pathway |

group ($P<0.01$) significantly reduced cerebral infarct volume in rats, suggesting butein can reduce cerebral damage caused by CI/R in a dose-dependent manner (Fig 14).

**3.8.5. LDH levels analysis.** Compared with the Sham group, LDH levels in the cerebral tissue and serum of rats in MCAO group was significantly increased ($P<0.01$), suggesting that MCAO caused cerebral damage. Compared with the MCAO group, butein low-dose ($P<0.05$, $P<0.01$), medium-dose ($P<0.01$), high-dose ($P<0.01$) and the Eda group ($P<0.01$) significantly reduced the LDH levels, suggesting that butein could inhibit the release of LDH, and showed a dose-dependent manner (Fig 15).

## 4. Discussion

IS is a major threaten to the human, however, there was no effective drugs used in clinical besides thrombolytic drugs. There are multiple biological processes in IS, including oxidative stress, inflammation, apoptosis, and energy metabolism disorders. A single drug may difficult to achieve therapeutic effects, drug cocktail therapy or TCM containing multiple compounds may be an effective treatment method.

DO has good effects in regulating qi and blood circulation, and is commonly used for treating IS. DO contains a large number of bioactive components, which are mainly flavonoids, including dihydroflavonoids, chalcone, isoflavones and so on. However, the effective substances and active mechanisms for the treatment of IS are still unclear. In this study, we

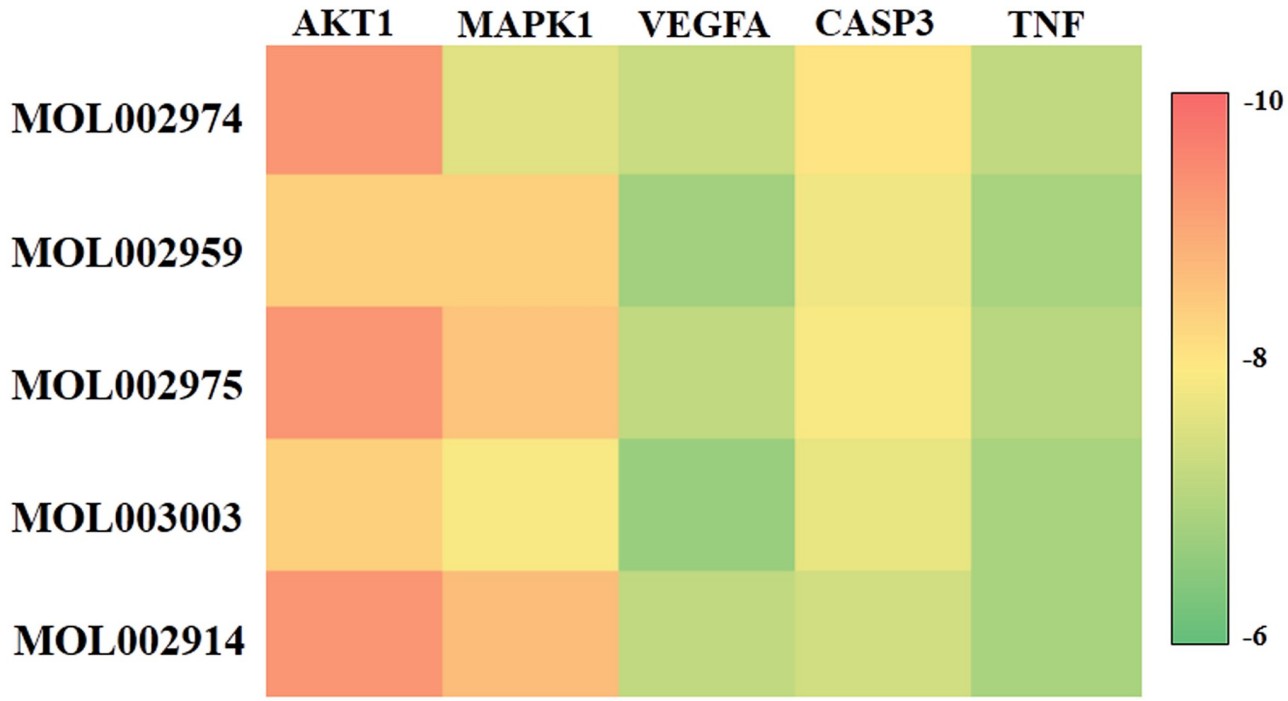

**Fig 8. Heat map of the difference of binding energy between potential bioactive components and top5 core targets.**

conducted component screening, target prediction and network analysis through network pharmacology to study the pharmacological mechanism related to DO and IS, which improved the accuracy of target prediction to a certain extent. Based on the analysis of the PPI system, it was found that 10 highly differentially expressed genes such as AKT1, MAPK1, VEGFA,

| PDB | AKT1 (3CQU) | Center[a] | X: 1.606 | Y: -0.784 | Z: 25.403 |
|---|---|---|---|---|---|
| | | Sizeb | X: 26.25 | Y: 22.5 | Z: 26.25 |
| **MOL ID** | MOL002974 | MOL002914 | MOL002975 | MOL002959 | MOL003003 |
| **Affinity (kcal/mol)** | -9.3 | -9.3 | -9.3 | -8.4 | -8.4 |
| **3D Structure** | | | | | |

a: Central coordinates of docking pockets, b: The size of the docking pockets in the x, y and z axis

**Fig 9. Optimal docking of potential bioactive components with top5 core targets.**

**Table 3. The binding site of the potential bioactive components and the core target protein AKT1.**

| Location | MOL002974 | MOL002914 | MOL002975 | MOL002959 | MOL003003 |
|---|---|---|---|---|---|
| THR211 | √ | | | | |
| THR291 | √ | √ | √ | | |
| ASN279 | √ | √ | | | |
| GLU278 | √ | | √ | | |
| PHE161 | √ | √ | √ | | √ |
| VAL164 | √ | √ | √ | √ | √ |
| GLU228 | √ | | | | |
| ALA230 | √ | √ | √ | | |
| GLU234 | √ | | | | |
| ALA177 | | √ | √ | √ | √ |
| LEU156 | | | √ | | |
| PHE438 | | | √ | | |
| ARG4 | | | | √ | |
| Number | 9 | 6 | 8 | 3 | 3 |

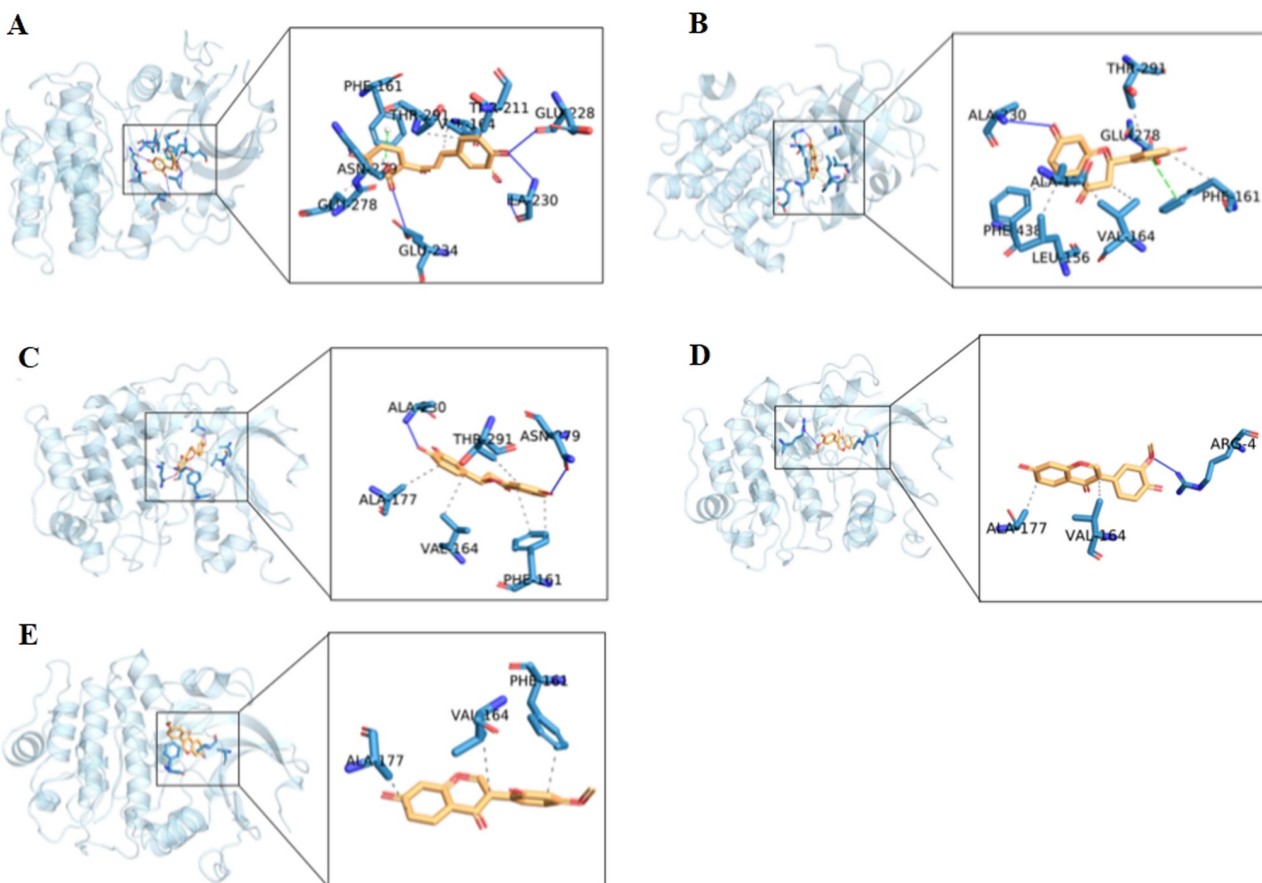

**Fig 10. The binding site of the potential bioactive components and the core target protein AKT1.** A:MOL002974, B:MOL002975, C:MOL002914, D:MOL002959, E:MOL003003. The blue solid lines represented hydrogen bondings, and the gray dashed lines represented hydrophobic interactions.

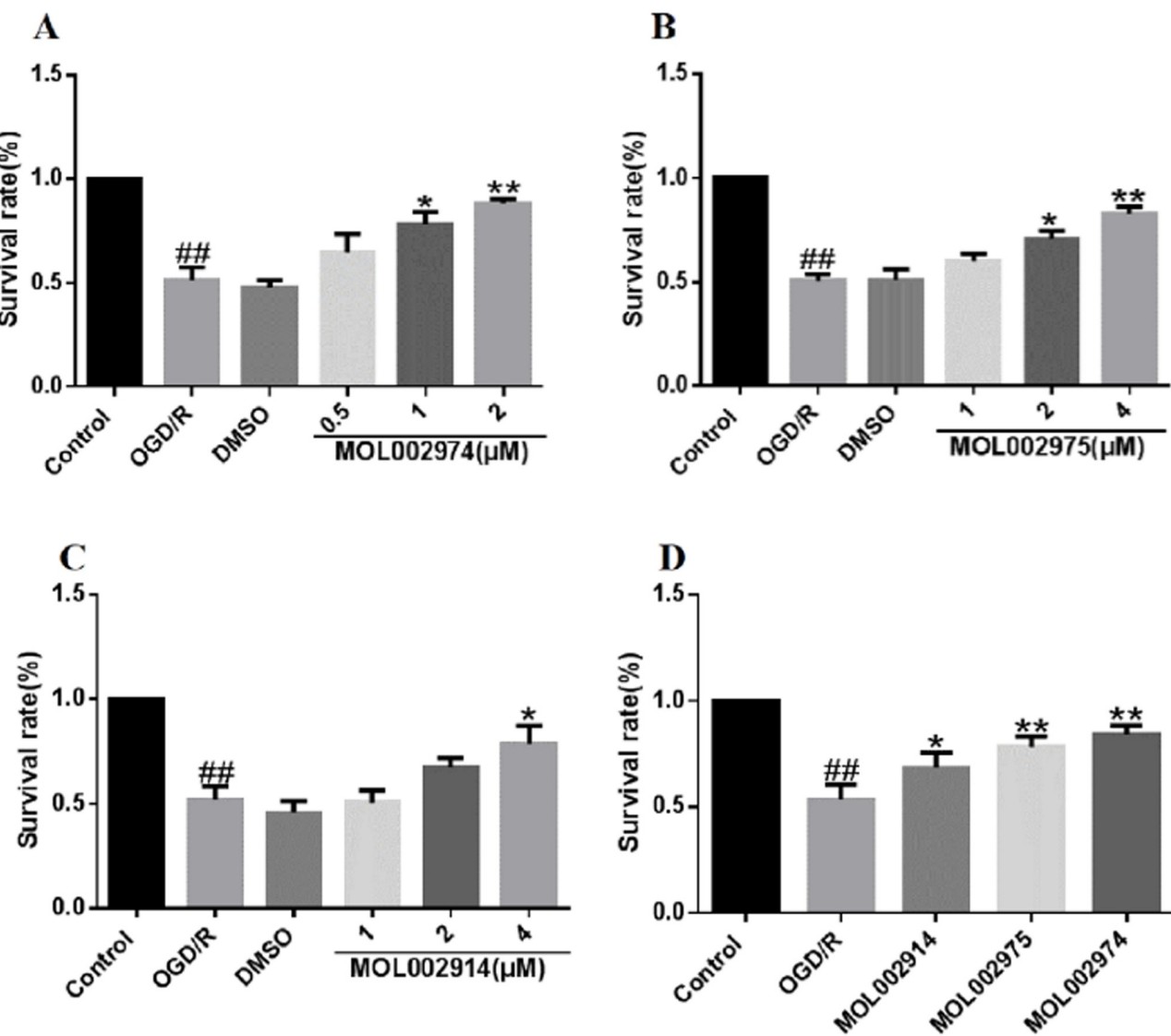

**Fig 11. Effects of 3 core bioactive components on the survival rate of PC12 cells.** A: MOL002974, B: MOL002975, C: MOL002914, D: 3 potential bioactive components. ## $P < 0.01$ vs control group; * $P < 0.05$, ** $P < 0.01$ vs OGD/R group. ($\bar{X} \pm$ SD, n = 3).

CASP3, TNF, MAPK8, PTGS2, STAT3, MMP9 and ESR1 played a key role in the pharmacological function of DO and were considered to be core targets. Network topology analysis showed that the process of DO affecting IS involved a variety of biological processes, cell composition and molecular functions, and was a complex process. At the same time, the core targets were significantly enriched in PI3K/Akt signaling pathway, TNF signaling pathway, MAPK signaling pathway, NF-κB signaling pathway, FoxO signaling pathway, Ras signaling pathway, Estrogen signaling pathway and other signaling pathways.

IS involved a variety of biological processes, including oxidative stress, inflammation and vascular endothelial regulation. Among them, oxidative stress was an important pathological link of nerve function injury under ischemia and hypoxia [21, 22], which was closely related to PI3K/Akt signaling pathway, FoxO signaling pathway, Estrogen signaling pathway, Ras

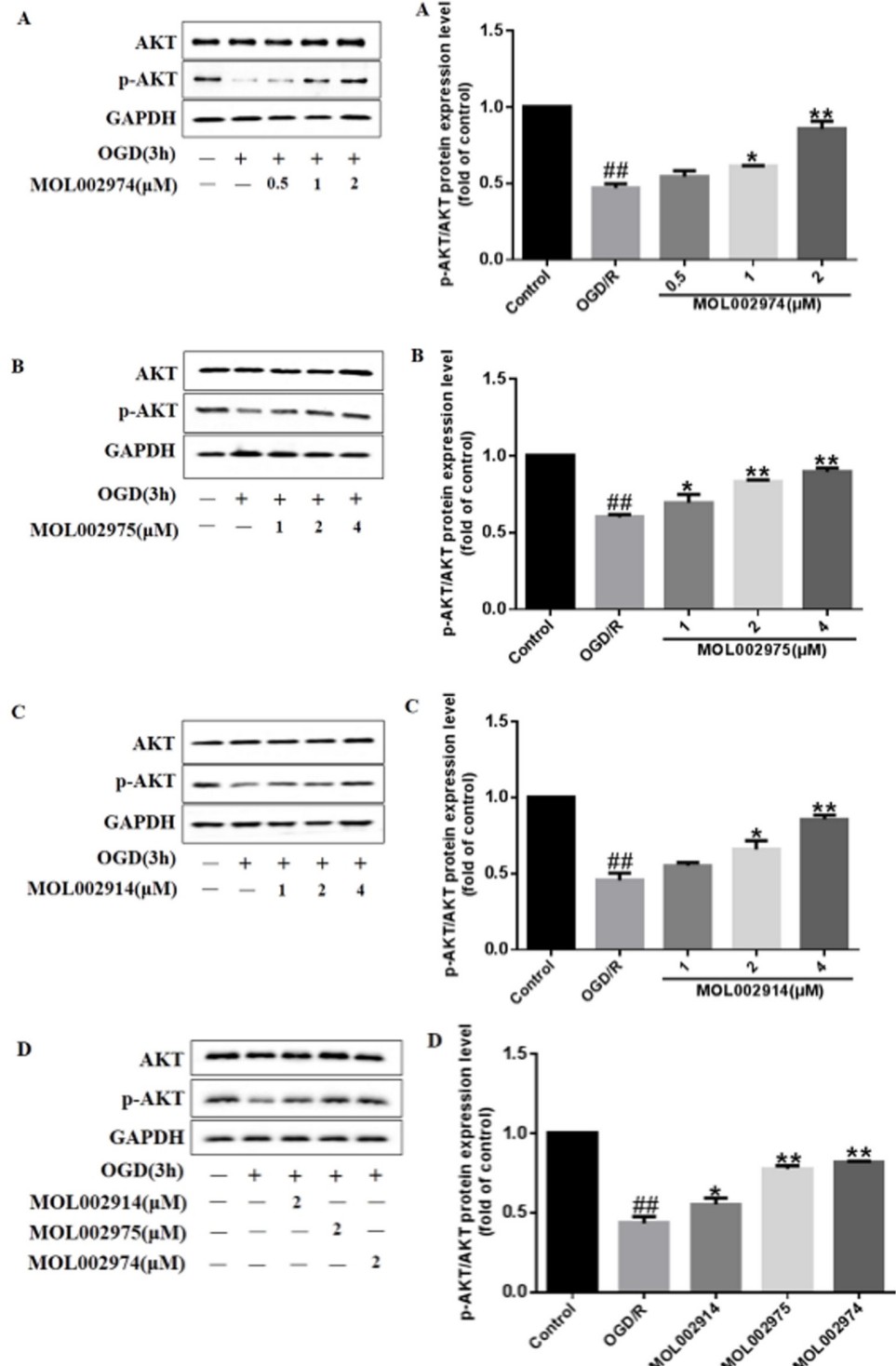

**Fig 12. Effects of core bioactive components on the core target protein expression of p-AKT/AKT induced by OGD/R in PC12 cells.** A: MOL002974, B: MOL002975, C: MOL002914, D: 3 potential bioactive components. Values are represented as mean ± SD from 3 independent experiments, each experiment repeated 3 times.[#]P<0.05 vs. Control group,[##]P<0.01 vs. Control group,[*]P<0.05 vs. OGD/R group, [**]P<0.01 vs. OGD/R group.

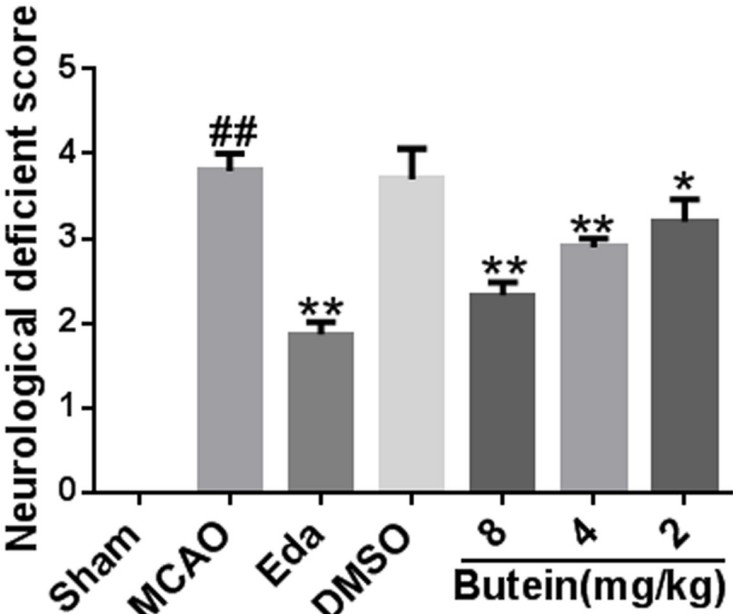

**Fig 13. Effects of butein on neurofunctional scores in MCAO rats.** $^{\#\#}$ $P<0.01$ vs Sham group; $^{*}P<0.05$, $^{**}P<0.01$ vs MCAO group. ($\bar{X} \pm$ SD, n = 6).

signaling pathway and other signaling pathways. PI3K/Akt was a classical signal transduction pathway that regulated cell survival, differentiation and apoptosis, and played an important biological role by regulating the downstream apoptosis-related proteins. A large number of studies had found that the PI3K/Akt pathway was a pro-survival signaling pathway, and the activation of the pathway helped to play a protective role on nerve cells, especially when the ischemic/hypoxic neurons were damaged [23–25]. The activated Akt could initiate the downstream cascade reaction of the PI3K/Akt signaling pathway, further phosphorylate a series of substrates, such as downstream Bad, Caspase-3, GSK-3β and exert its function of regulating cell differentiation, promoting cell survival and anti-apoptosis through various channels [26]. Studies had shown that [27] activating the PI3K/Akt signaling pathway could inhibit the

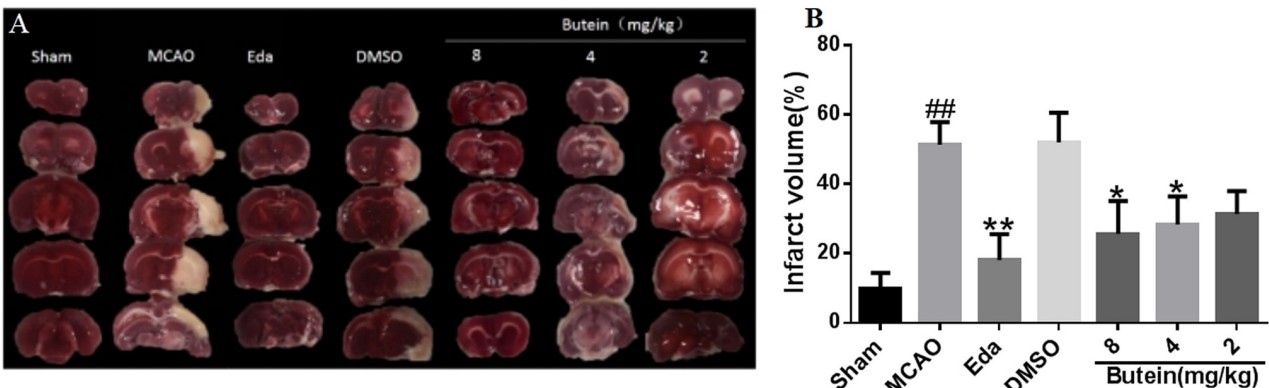

**Fig 14. Effects of butein on cerebral infarct volume of MCAO rat.** A: TTC stained cerebral sections of each group; B: Percentage of TTC stained infarct volume. $^{\#\#}$ $P<0.01$ vs Sham group; $^{*}P<0.05$, $^{**}P<0.01$ vs MCAO group. ($\bar{X} \pm$ SD, n = 3).

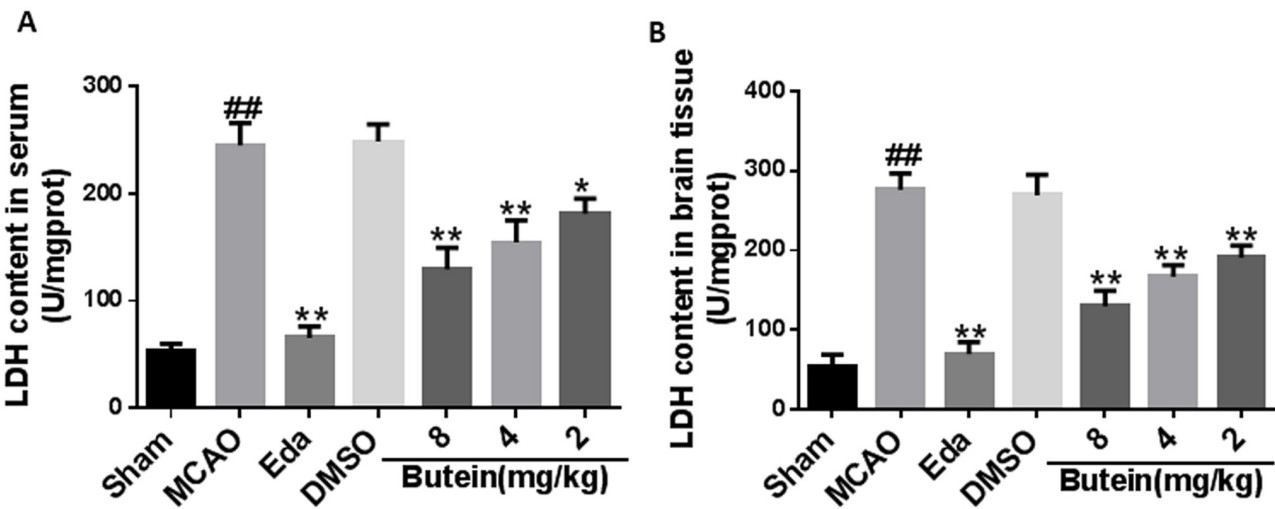

**Fig 15. Effects of butein on LDH levels in MCAO rats.** A: Effects of butein on LDH level in serum of MCAO rats. B: Effects of butein on cerebral tissue LDH level of MCAO rats. $^{##}$ $P<0.01$ vs Sham group; $^{*}P<0.05$, $^{**}P<0.01$ vs MCAO group. ($\bar{X} \pm$ SD, n = 3).

activation of Caspase-3 to play a central neuroprotective effect. In addition, studies had found that [28] estradiol could increase cell viability, reduce the production of Reactive oxygen species (ROS), activate Akt signal and inhibit GSK-3β involved in neurodegenerative changes, promote the separation of Nrf2 from Keap1, and significantly increase HO-1 expression and SOD activity. The inflammatory response ran through the whole process of the occurrence and development of IS.

After cerebral ischemia, extensive brain tissue necrosis occurred in the ischemic region because of energy depletion, which released a large number of inflammatory mediators, thus activating the immune response and further promoting the release of inflammatory factors [29]. It was closely related to TNF signaling pathway, NF-κB signaling pathway, MAPK signaling pathway and other signaling pathways. MAPK was an important transmitter of signals from the cell surface to the inside of the nucleus. They regulated many physiological activities, such as inflammation and apoptosis [30]. MAPK activated pro-inflammatory factors such as TNF-α, interleukin family (IL1, IL6, etc.) to exacerbate the inflammatory response [31]. TNF-α was a pro-inflammatory factor with multiple pro-inflammatory and neurotoxic effects. It was the initiating factor of inflammatory response and had complex biological activities. During the early stage of cerebral ischemia, increased TNF-α secretion or synthesis was the main cause of cerebral infarction [32]. Experiments had shown that an increase in the amount of TNF-α could promote the inflammatory response after cerebral ischemia/reperfusion (CI/R) and aggravated brain damage [33], while TNF-α inhibitors could reduce CI/R injury [34]. In addition, studies had shown that inhibiting the expression of NF-κB could reduce the cerebral infarction area and neuronal death in MCAO rats [35]. The function of vascular endothelial regulation was closely related to VEGF signaling pathway and HIF-1 signaling pathway. HIF-1α was a hypoxia-induced nuclear transcription factor. Activated HIF-1α under ischemia-hypoxia state was activated, induced the transcriptional expression of downstream gene VEGF, participated in angiogenesis, and regulated cell adaptation to hypoxia [36].

Moreover, 5 potential bioactive components with degree≥20 among the 12 potential bioactive components were selected to be docked with the top5 core targets using Autodock Vina software to further analyze the possible core bioactive components that affected the IS

according to the relationship between the potential components and the core target proteins. The results showed that the 5 potential bioactive components could bind stably with the top5 core targets which compared with the positive drugs, and the binding energy is lower, and the binding is more stable. It showed that in the process of DO affecting IS, these 5 potential bioactive components played an important role by regulating top5 core targets. Meanwhile, the 5 potential bioactive components showed better binding to AKT1 target, followed by MAPK1, CASP3, TNF and VEGFA. Based on the docking results, we focused on analyzing the binding sites of MOL002974, MOL002914 and MOL002975 with AKT1. In addition, the CCK8 method was used to detect the effects of 3 core bioactive components on the cell viability of PC12 cells. Western blot experiments were used to verify the regulatory effects of the 3 core bioactive components on the AKT of PC12 cells.

The cell viability results showed that MOL002974, MOL002975, and MOL002914 all improved the cell survival rate in a dose-dependent manner and alleviated the damage to PC12 cells in the OGD/R group, suggesting that the 3 core bioactive components could all promote cell survival. In addition, MOL002974 had the best effect on improving cell survival when the 3 core bioactive components are at the same concentration, suggesting that MOL002974 played a more important role in improving cell survival than the other 2 core bioactive components.

The results showed that, compared with the control group (no treatment), the OGD/R group inhibited the phosphorylation of AKT ($p < 0.01$). However, 3 core bioactive component treatments significantly up-regulated the phosphorylation of AKT in a dose dependent manner ($p < 0.01$), respectively. In addition, 3 core bioactive components could up-regulate AKT phosphorylation under the same conditions, and MOL002974 had a slightly better effect.

The results of *in vitro* experiments showed that OGD/R could inhibit cell survival and AKT phosphorylation which were reversed by the 3 core bioactive components. Among them, MOL002974 (butein) had a slightly better effect. Therefore, the protective effect of MOL002974 (butein) against cerebral ischemia was further evaluated in a rat model of middle cerebral artery occlusion (MCAO) by detecting neurological score, cerebral infarction volume and lactate dehydrogenase (LDH) level. The results indicated that MOL002974 (butein) could significantly improve the neurological score of rats, decrease cerebral infarction volume, and inhibit the level of LDH in the cerebral tissue and serum in a dose-dependent manner.

The binding sites of MOL002974, MOL002975, MOL002914 and the target protein AKT1 were 9, 8, and 6, respectively. Some of the binding sites were consistent, which may be the reason for the similar docking results of the 3 core bioactive components. Akt called protein kinase B (PKB), was a serine/threonine kinase that could be activated by catalyzing the phosphorylation of its own serine and threonine sites. MOL002974, MOL002975, MOL002914 and AKT1 binding sites had 2, 1, and 0 threonine residues, respectively. Therefore, we assumed that the more threonine residues in the binding site of AKT1, the bioactive components activation effect were better. Further experimental results showed that MOL002974 had the best effect on up-regulating AKT phosphorylation, which was in preliminary agreement with our hypothesis. The specific mechanism of action needs further experimental study and verification.

## 5. Conclusions

In summary, the results indicated that the bioactive components of DO may affect IS through important signaling pathways in the biological process of "oxidative stress", "inflammatory response" and "vascular endothelial function regulation", such as PI3K/Akt signaling pathway, TNF signaling pathway, MAPK signaling pathway. And through PPI network analysis, it was

determined that 10 core targets including AKT1, MAPK1, VEGFA, CASP3, TNF, MAPK8, PTGS2, STAT3, MMP9 and ESR1, participated in these processes. The 5 potential bioactive components were docked with the top5 core targets by molecular docking software, and the 3 core bioactive components affecting the IS were further analyzed and verified by experiments. To a certain extent, this research reveals the potential mechanism of DO affecting IS, and provides a basis for the secondary development of DO.

## Supporting information

**S1 Raw images.**
(RAR)

**S1 Data.**
(RAR)

## Author Contributions

**Conceptualization:** Sha Chen, Taiwei Dong.

**Data curation:** Xinming Lu, Jing Li.

**Funding acquisition:** Miaomiao Xi.

**Investigation:** Kedi Liu, Xingru Tao, Jing Su.

**Methodology:** Kedi Liu, Xingru Tao, Jing Su.

**Project administration:** Miaomiao Xi.

**Resources:** Fei Li, Fei Mu, Shi Zhao.

**Software:** Fei Li, Fei Mu, Shi Zhao.

**Writing – review & editing:** Kedi Liu, Xingru Tao, Jing Su, Jialin Duan, Peifeng Wei, Miaomiao Xi.

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
