## [Decision Letter · Decision Letter 0]

8 Apr 2021

PONE-D-21-05118

Network pharmacology and molecular docking reveal the effective substances and active mechanisms  of Dalbergia Odorifera in protecting against ischemic stroke

PLOS ONE

Dear Dr. xi,

Thank you for submitting your manuscript to PLOS ONE. After careful consideration, we feel that it has merit but does not fully meet PLOS ONE’s publication criteria as it currently stands. Therefore, we invite you to submit a revised version of the manuscript that addresses the points raised during the review process.

We look forward to receiving your revised manuscript.

Kind regards,

Lucio Annunziato, MD

Academic Editor

PLOS ONE

Journal Requirements:

Reviewers' comments:

Reviewer's Responses to Questions

**Comments to the Author**

1. Is the manuscript technically sound, and do the data support the conclusions?

Reviewer #1: Yes

Reviewer #2: Yes

2. Has the statistical analysis been performed appropriately and rigorously? 

Reviewer #1: Yes

Reviewer #2: Yes

3. Have the authors made all data underlying the findings in their manuscript fully available?

Reviewer #1: Yes

Reviewer #2: Yes

4. Is the manuscript presented in an intelligible fashion and written in standard English?

Reviewer #1: Yes

Reviewer #2: Yes

5. Review Comments to the Author

Reviewer #1: The paper by Kedi Liu and collegues try to elucidate the mechanism of action of Dalbergia Odorifera (DO), a species of legume in the family Fabacae, endemic to China, as a phytocomplex able to promote circulation and removing blood stasis in stroke, since it is widely used according to what the authors say for clinical treatment of cardiovascular and cerebrovascular diseases.

To this purpose authors analyzed the effective substances contained in DO in order to identify those responsible of the pharmacological actions observed, among flavonoids and volatile oils of which the plant is rich. To do that, authors implemented a network pharmacology approach aimed at a first phase to a screening of bioactive components of DO, a target prediction of bioactive compounds, then to a network construction based on Ischemia related targets followed by an enrichment analysis followed and a molecular docking confirmation then, finally to an experimental validation of molecular targets identified in PC12 cells subjected to OGD. By western blotting experiments Kedi and collaborators found that 5 potential bioactive components of DO strongly bind to the target protein AKT1.

Although the present research work is innovative from the point of view of network pharmacology, as it goes beyond the traditional approach based on extractive chemistry, it is well known from the literature that PI3K/Akt pathway constitutes a pro-survival signaling pathway, and the activation of that pathway helps to play a protective role on nerve cells, especially when the ischemic/hypoxic neurons were damaged.

The present paper will be certainly more exhaustive for the audience of readers of this magazine if the authors will confirm, in an adequate in vivo model of cerebral ischemia, the efficacy and the effective dosage of the extracted drugs from Dalbergia Odorifera.

Reviewer #2: The paper by Kedi Liu and colleagues investigates the effective components and possible mechanisms of action of Dalbergia Odorifera (DO) in ischemic stroke. In the first part of study, they collected 98 components of DO by using TCMSP, and 28 bioactive components were selected with the screening conditions of OB≥40% and DL≥0.18. Then, the authors performed the predict analysis of target through the Swiss Target Prediction database, obtaining a total of 544 monomer component targets. The component-target network was constructed by using Cytoscape software, which contained 572 nodes and 2772 edges. By using three different database (NCBI, CTD, and GeneCards) they selected 344 targets involved in the development to ischemic stroke. 544 monomer component targets and 344 disease targets were intersected by using Venny online drawing tool, and a total of 71 common targets were obtained. Subsequently, researchers identified 101 nodes and 498 edges with Cytoscape software, that elaborated a complex network based on the interactions among bioactive components, targets and the disease. In addition, in order to predict functional interactions of proteins, they used the string tool to acquire Protein-protein interaction (PPI) network for the 71 overlapped targets with a combined score greater than 0.4 and ‘Homo sapines’ as selecting criterions, the network of PPI consisted of 71 nodes and 815 edges. The network of PPI was elaborated by cytoHubba plug-ins, and 10 main targets were identified (AKT1, MAPK1, VEGFA, CASP3, TNF, MAPK8, PTGS2, STAT3, MMP9 and ESR1). Twelve potential bioactive components have been screened and it has been showed that these components were flavonoids, including dihydroflavonoids and isoflavones. Furthermore, it has been performed GO functional enrichment analysis and KEGG pathway enrichment in DAVID, identifying 148 biological processes (BPs), 16 cell compositions (CCs), and 18 molecular functions (MFs), accounting for 81%, 9%, and 10%, respectively. These biological processes have been divided into 3 parts, including oxidative stress, inflammatory response and regulation of vascular endothelial function. In order to evaluated the core bioactive components of DO affecting on ischemic stroke, it has been analyzed the affinity of 5 potential bioactive components and the following top5 core target proteins: AKT1 (PDB: 3CQU), MAPK1 (PDB: 5K4I), VEGFA (PDB: 3BDY), CASP3 (PDB: 3H0E) and TNF (PDB: 4TWT), respectively. The data demonstrated that the 5 potential bioactive components could bind to the top5 targets and strongly bind to the target protein AKT1, the binding of each small molecule to the target protein mainly depended on hydrophobic interaction and hydrogen bonding. In the last part of study the authors confirmed that components MOL002974, MOL002975 and MOL002914 had as target AKT. The data showed the 3 core bioactive component treatments significantly up-regulated the phosphorylation of AKT in a dose dependent manner. In addition, 3 core bioactive components could up-regulate AKT phosphorylation and in particular MOL002974 had a slightly better effect.

Although the manuscript technically sounds, experiments have been performed with rigor, through appropriate controls, replication and sample size and the data produced support the conclusions, some points should be improved to reinforce the significance of the paper.

• In ”methods” section, it is necessary to explain the rationale of dose choose of MOL002974, MOL002975, and MOL002914 providing information relating to preliminary data.

• The authors demonstrated that there was a significant increase of p-AKT expression, after treatment, for 24 hours, of PC12 cells incubated with MOL002974, MOL002975, and MOL002914. It would be interesting to show the effects of these components on cell survival.

6. PLOS authors have the option to publish the peer review history of their article (what does this mean?). If published, this will include your full peer review and any attached files.

Reviewer #1: **Yes: **Antonio Vinciguerra

Reviewer #2: No

---

## [Author Response · Author response to Decision Letter 0]

13 Jul 2021

Replies to Reviewer 1

Q: The present paper will be certainly more exhaustive for the audience of readers of this magazine if the authors will confirm, in an adequate in vivo model of cerebral ischemia, the efficacy and the effective dosage of the extracted drugs from Dalbergia Odorifera.

A: Thank you for your suggestions. MOL002974 (butein) isolated from Dalbergia odorifera is an important core bioactive component screened by network pharmacology and molecular docking. In order to evaluate the cerebral protection of butein, we replicated the middle cerebral artery occlusion (MCAO) model in rats to detect the effects of different doses of MOL002974 (butein) on neurological score, infarct volume and lactate dehydrogenase (LDH) level of MCAO rats. The results indicated that MOL002974 (butein) could significantly improve the neurological score of rats, decrease cerebral infarction volume, and inhibit the level of LDH in a dose-dependent manner. Details have been added to the manuscript.

Replies to Reviewer 2

Q1: In ”methods” section, it is necessary to explain the rationale of dose choose of MOL002974, MOL002975, and MOL002914 providing information relating to preliminary data.

A: Thanks for your constructive comments. In order to choose the appropriate dosage, CCK8 method was used to detect the effects of different concentrations of 3 core bioactive components (MOL002974, MOL002975 and MOL002914) on the survival rate of PC12 cells of oxygen glucose deprivation/reperfusion (OGD/R). The results have been submitted as supplementary material.

Q2: The authors demonstrated that there was a significant increase of p-AKT expression, after treatment, for 24 hours, of PC12 cells incubated with MOL002974, MOL002975, and MOL002914. It would be interesting to show the effects of these components on cell survival.

A: Thank you for your suggestions. The CCK8 method was used to detect the effects of 3 core bioactive components on cell viability. The cell viability results showed that MOL002974, MOL002975, and MOL002914 all improved the cell survival rate in a dose-dependent manner and alleviated the damage to PC12 cells in the OGD/R group, suggesting that the 3 core bioactive components could all promote cell survival. In addition, MOL002974 had the best effect on improving cell survival when the 3 core bioactive components are at the same concentration, suggesting that MOL002974 played a more important role in improving cell survival than the other 2 core bioactive components. Details have been added to the manuscript.

---

## [Decision Letter · Decision Letter 1]

23 Jul 2021

Network pharmacology and molecular docking reveal the effective substances and active mechanisms  of Dalbergia Odorifera in protecting against ischemic stroke

PONE-D-21-05118R1

Dear Dr. xi,

We’re pleased to inform you that your manuscript has been judged scientifically suitable for publication and will be formally accepted for publication once it meets all outstanding technical requirements.

Kind regards,

Lucio Annunziato, MD

Academic Editor

PLOS ONE

Additional Editor Comments (optional):

Reviewers' comments:

Reviewer's Responses to Questions

**Comments to the Author**

1. If the authors have adequately addressed your comments raised in a previous round of review and you feel that this manuscript is now acceptable for publication, you may indicate that here to bypass the “Comments to the Author” section, enter your conflict of interest statement in the “Confidential to Editor” section, and submit your "Accept" recommendation.

Reviewer #1: All comments have been addressed

Reviewer #2: (No Response)

2. Is the manuscript technically sound, and do the data support the conclusions?

Reviewer #1: Yes

Reviewer #2: (No Response)

3. Has the statistical analysis been performed appropriately and rigorously? 

Reviewer #1: Yes

Reviewer #2: (No Response)

4. Have the authors made all data underlying the findings in their manuscript fully available?

Reviewer #1: Yes

Reviewer #2: (No Response)

5. Is the manuscript presented in an intelligible fashion and written in standard English?

Reviewer #1: Yes

Reviewer #2: (No Response)

6. Review Comments to the Author

Reviewer #1: Authors accepted the request for further confirmatory experiments in vivo so that, in its current form, this manuscript is certainly considerable for publication.

Reviewer #2: (No Response)

7. PLOS authors have the option to publish the peer review history of their article (what does this mean?). If published, this will include your full peer review and any attached files.

Reviewer #1: **Yes: **Antonio Vinciguerra

Reviewer #2: No

---

## [Editor Report · Acceptance letter]

16 Sep 2021

PONE-D-21-05118R1 

Network pharmacology and molecular docking reveal the effective substances and active mechanisms of Dalbergia Odoriferain protecting againstischemic stroke 

Dear Dr. Xi:

I'm pleased to inform you that your manuscript has been deemed suitable for publication in PLOS ONE. Congratulations! Your manuscript is now with our production department. 

Kind regards, 

on behalf of

Dr. Lucio Annunziato 

Academic Editor

PLOS ONE